# Total Knee Arthroplasty in Unrecognized Septic Arthritis—A Descriptive Case Series Study

**DOI:** 10.3390/antibiotics12071153

**Published:** 2023-07-06

**Authors:** Florian Hubert Sax, Bernd Fink

**Affiliations:** 1Department of Joint Replacement, General and Rheumatic Orthopaedics, Orthopaedic Clinic Markgröningen gGmbH, Kurt-Lindemann-Weg 10, 71706 Markgröningen, Germany; florian.sax@rkh-gesundheit.de; 2Orthopaedic Department, University Hospital Hamburg-Eppendorf, Martinistrasse 52, 20246 Hamburg, Germany

**Keywords:** total knee arthroplasty, septic arthritis, periprosthetic joint infection, Copal G+C cement

## Abstract

Background: Synovitis, like that associated with chronic bacterial arthritis, is a very rare finding during the implantation of knee endoprostheses. In such cases, we fix the knee prostheses with cement containing two antibiotics and carry out a course of systemic antibiotic administration. The aim was to analyze these cases for incidence, detection of bacteria, risk factors, and outcome. Methods: Out of 7534 knee replacements between January 2013 and December 2020, 25 cases were suspected during the surgical procedure to have suffered from bacterial arthritis and were treated accordingly. Total synovectomy was carried out, whereby five intraoperative synovial samples were examined bacteriologically, and the complete synovitis was analyzed histologically. The mean follow-up was 65.3 ± 27.1 (24–85) months. Results: In nine cases (0.12%), the diagnosis of bacterial arthritis was made histologically and by clinical chemistry (elevated CRP), and in two of these cases, pathogen verification was performed. Eight of these nine patients had previously had injections or surgery associated with the corresponding knee joint or had an underlying immunomodulatory disease. None of the patients developed a periprosthetic infection at a later stage. Conclusion: With an incidence of 0.12%, it is rare to unexpectedly detect bacterial synovitis during surgery. Total synovectomy, use of bone cement with two antibiotics, and immediate systemic antibiotic therapy seem to keep the risk of periprosthetic infection low.

## 1. Introduction

Total knee arthroplasty (TKA) represents a proven treatment method for patients with primary or secondary osteoarthritis of the knee joint [1]. The knee usually shows signs of synovitis at the time of implantation [2,3]. Very rarely, synovitis is observed intraoperatively that macroscopically suggests subacute or chronic bacterial arthritis without the preoperative clinical and laboratory diagnostic tests having indicated this [4]. This bacterial arthritis is not preoperatively recognized because the clinical symptoms are not indicative, and laboratory parameters like C-reactive protein (CRP) are not, or only slightly, elevated [5,6]. It is unclear how often such unexpected, intraoperatively observed bacterial synovitis occurs. There have not been any numbers or percentages reported in the medical literature. Moreover, there are no papers in the literature that concern the therapy and results of such cases. Therefore, it is unclear how to proceed surgically in such a case: whether total knee arthroplasty can be performed directly after a total synovectomy or if insertion of a spacer in the first step should be preferred, followed by implantation of a total knee prosthesis in a second step.

In such cases, where we observe synovitis during surgery that is macroscopically suggestive of bacterial arthritis but without preoperative clinical symptoms, our treatment protocol is derived from the treatment of total knee replacement following a septic arthritis history and septic one-stage revision arthroplasty of infected total knee arthroplasties [5,6,7]: After total synovectomy, total knee arthroplasty is performed with bone cement containing two antibiotics (Copal G+C, Heraeus Medical, Werheim, Germany). The patients are then given systemic antibiotic therapy until the histological and microbiological results of the samples taken intraoperatively are available. In the event of histological and/or microbiological evidence of bacterial arthritis, systemic antibiotic therapy is continued intravenously for a total of 2 weeks and orally for 4 weeks. As soon as such bacterial arthritis is ruled out, antibiotic treatment is discontinued.

It is still unknown whether the implantation of a knee endoprosthesis in cases of intraoperatively suspected, subacute, or chronic bacterial arthritis leads to an increased rate of periprosthetic infections of TKA. Implantation of total knee endoprostheses after previously occurring bacterial arthritis is generally associated with a significantly increased risk of periprosthetic joint infection (PJI) [5,8,9,10]. While the rate of periprosthetic infections after total knee arthroplasty is usually around 1 to 2% [6,7,8], Seo et al. [10] found an infection rate of 9.7% in 62 total knee arthroplasties after previously occurring bacterial arthritis, and Bauer et al. [5] found a PJI-rate of 9.4% in 53 patients with arthroplasties of 31 knees and 22 hips after bacterial arthritis. Bettencourt et al. [8] saw a 6.1-fold increased risk of periprosthetic infection of a total knee arthroplasty after bacterial arthritis in a case–control study compared to patients with gonarthrosis alone. In contrast, Ohlmeier et al. [11] found a lower infection rate of 2.9% in 68 cases after implantation of a total knee endoprosthesis when systemic antibiotic therapy was given at the same time.

To the best of our knowledge, there are no publications about knee arthroplasties in cases of intraoperatively suspected bacterial arthritis to date. Therefore, we evaluated our database for such cases in order to determine the following:The frequency of intraoperatively suspected bacterial arthritisThe number of confirmed bacterial arthritidesThe number of patients with bacterial arthritis where pathogens could be successfully isolatedThe risk factors of these patients with bacterial arthritisThe infection rate after implantation of total knee arthroplasty in these cases of confirmed bacterial arthritis

## 2. Results

Of the 7534 total knee arthroplasties (TKAs) in our single-center study, subacute or chronic bacterial arthritis was suspected intraoperatively in 25 cases (0.33%) because of the macroscopic appearance of the synovitis. In 20 cases, the synovium was found intraoperatively to be very thickened, and there were adhesions with pouch formation (stage III of septic arthritis according to Gächter and Stutz [4]); in five cases, there was pannus formation with infiltration into the cartilage (stage IV of septic arthritis according to Gächter and Stutz [4]). In none of these cases was there a preoperative clinical suspicion of bacterial arthritis: no hyperthermia, no reddening of the knee joint, no leucocyte elevation in the blood, and only a discrete increase in C-reactive protein (CRP) (Table 1)

Of the 25 intraoperative suspected synovitis cases of the 7534 knee replacements (0.33%), 9 (0.12%) were classified as bacterial (Table 1). In these cases, a pathogen was detected twice (once *Cutibacterium acnes* in 5 of 5 samples and once *Staphylococcus saprophyticus* in 3 of 5 samples) (Table 1 and Table 2). In three of these patients, the joint had been injected with hyaluronic acid shortly beforehand, and in three cases, the knee had already been operated on (two cases of arthroscopy, one of osteosynthesis) (Table 1). Two patients suffered from rheumatoid arthritis, and one patient from uterine and thyroid cancer (Table 1).

The Knee Society Score [12] at the two-year follow-up of the 9 patients was 86.1 ± 9.9 (70–100) points. No cases of periprosthetic infection of the total knee prosthesis or other complications were observed at the follow-up.

## 3. Discussion

To the best of our knowledge, it is not known how often bacterial arthritis is unexpectedly present during the implantation of a total knee endoprosthesis or how to proceed in the event of such an intraoperative finding. Can a total knee arthroplasty be implanted after a total synovectomy, or is it better to implant a mobile spacer first and implant the knee endoprosthesis in a second step? The incidence of such unexpected bacterial arthritis during implantation of a total knee endoprosthesis was very low at 0.12% (9 of 7534 total knee arthroplasties) in our study. The fact that in none of our cases, a periprosthetic infection occurred after implantation of a total knee endoprosthesis seems to indicate that, under certain conditions, total knee arthroplasty combined with local and systemic antibiotic administration can be performed in such cases.

Firstly, in our opinion, a case of acute bacterial arthritis with local and systemic signs of infection has to be ruled out and should not be treated in this manner. The cases in our study were not manifest, acute arthritides with clear clinical symptoms; they were subacute or chronic arthritides with minor signs of inflammation (only slightly elevated CRP levels) and with evidence of pathogens in only two cases. Therefore, most of the infections in our study remained culture-negative with no evidence of a microorganism (7 out of 9). It is not uncommon that bacteria cannot be cultured from subacute and chronic septic arthritis. Paz et al. [13] found no pathogens in 85 cases (27.8%) of 306 adult patients with septic arthritis and Chuckpaiwong and Phoompoung [14] in 36 of 62 patients (58.1%) with subacute or chronic septic arthritis. Culture-negative septic arthritides have significantly better outcomes than those where pathogens are detected, according to the study by Paz et al. [13].

Secondly, all our cases were implanted with a bone cement (Copal G+C, Heraeus Medical, Werheim, Germany) that releases two antibiotics (Gentamycin and Clindamycin). The use of two antibiotics in the bone cement for fixation of the total knee arthroplasty leads to a synergistic effect, and the elution of the individual components is better than that of the individual antibiotic alone in a cement with only one antibiotic [15,16]. In addition, our patients received systemic antibiotic therapy directly, which was initiated with a broad-spectrum antibiotic, and then, if the pathogen was identified, the antibiotic was specified for its susceptibility. The positive effect of such systemic antibiotic therapy is supported by the fact that Ohlmeier et al. [11] found a lower infection rate of 2.9% in 68 patients with a history of septic arthritis after implantation of a total knee arthroplasty and systemic antibiotic therapy, compared to around 10% in the studies by Seo et al. [10], Bauer et al. [5] and Bettencourt et al. [8] without antibiotic therapy.

Eight of the nine patients in the current study had had injections into the corresponding knee joint, previous operations, or an underlying immune-modulating disease. Two patients had rheumatoid arthritis, for which an increased risk for bacterial arthritis and post-operative risk of infection is known [17,18,19,20,21,22,23,24]. In particular, the incidence of septic arthritis is approximately 10-fold higher in patients with RA in comparison to the general population. The higher risk is related to the immune-modulating disease RA and the immune-suppressive drugs for this disease [23,24]. According to the studies of Bozic et al. [25] and Laupland et al. [26], a tumor in the patient’s history, as in the case of one patient in our study, also indicates an increased risk of septic arthritis and periprosthetic infections after implantation of a TKA. Previous surgery, which was performed on the same knee in 3 of the patients in our study, also represents a known risk factor for bacterial arthritis and periprosthetic infection [25,27,28,29]. Additionally, previous intra-articular injections with corticoids or hyaluronic acid also significantly increase the risk of septic arthritis and periprosthetic infections after the implantation of knee prostheses [30,31,32,33]. Previous injections with hyaluronic acid had been performed in 3 patients in our study.

The study has some limitations. The small number of cases is due to the very rare situation where bacterial arthritis and synovitis are unexpectedly discovered intraoperatively and the single-center study design. The retrospective study design is also a consequence of the rarity of the event. The study quality could be improved by a future prospective multicenter study. In addition, it cannot be completely ruled out that a periprosthetic infection may occur at a later time in our cases, although the fact that periprosthetic reinfections after septic revisions usually occur within 2 years after surgery argues against this [34]. Polymorphonuclear leukocytes are also occasionally seen in the synovial membrane of patients with rheumatoid arthritis without the presence of joint infection [35,36]. However, the relatively high number of neutrophil leukocytes of 20 per high-power field in the two cases with rheumatoid arthritis in our study also suggested, in our opinion, a subacute or previous bacterial infection of the joint. Other joint diseases that might explain neutrophilic granulocytes in the synovial membrane were not diagnosed in the patients of this study. However, as a further limitation, it must be mentioned that there is no classification system for bacterial arthritis that gives clear cut-off values for the CRP level or for the number of neutrophils per high power field above which sub-acute or chronic septic arthritis can be diagnosed. Therefore, in this study, we have used the classification for bacterial periprosthetic infections [37,38,39,40,41]. Additionally, because of the low number of patients, a statistical analysis for analyzing risk factors in comparison to the whole patient group without bacterial arthritis (7525 cases) could not be done. Therefore, only a descriptive study without statistical tests was done. In the future, multicenter studies with more patients are needed to obtain additional information about the incidence, risk factors, and results of treatment of these rare, unexpected bacterial arthritis during implantation of TKA.

## 4. Materials and Methods

Between January 2013 and December 2020, 7534 total knee arthroplasties (TKA) were carried out in our clinic and collected in our clinic’s database. All patients were routinely tested preoperatively for leukocyte count and C-reactive protein (CRP) levels in the blood. Of these, bacterial arthritis was suspected intraoperatively in 25 cases (0.33%) because of the macroscopic appearance of the synovitis. Because of the unexpected intraoperative suspicion of bacterial synovitis, the total knee arthroplasty was performed after a complete synovectomy and fixation of the prosthesis with a bone cement that releases two antibiotics (Gentamycin and Clindamycin from the Copal G+C cement, Heraeus Medical, Werheim, Germany). Systemic antibiotic therapy was then initiated with a broad-spectrum antibiotic (2nd generation Cephalosporin). The patient cohort consisted of 10 women and 15 men aged 70.3 ± 5.1 (mean, standard deviation) (55–85) years. These cases were registered in the database and were routinely followed up for at least 2 years in order to monitor the clinical outcomes of these patients. The data were evaluated retrospectively. The study has been performed in accordance with the PROCESS 2020 reporting guidelines for case series [42].

During surgery, five samples of the synovial tissue were obtained from synovitis and were immediately transferred to the microbiology unit. The remaining synovial tissue was submitted for histological examination.

Immediately upon arrival in the microbiology unit, synovial samples were minced under sterile conditions and incubated on chocolate agar, Columbia agar, McConkey agar, and Schaedler agar under aerobic conditions for 2–3 days and anaerobic conditions for 5 days. In addition, two aliquots were incubated in BHI and in thioglycollate broth for two weeks, as previously described [37,43,44,45,46]. Media were checked daily for bacterial growth. Broths that remained clear were incubated for 14 days until the specimen was declared negative, as described by Steinbrink and Frommelt [43], Ince et al. [38], and Schäfer et al. [44]. Subcultivation of turbid BHI broth was performed on Columbia, MacConkey, and chocolate agar in order to provide optimal growth conditions for fastidious and nonfastidious organisms. Turbid thioglycolate broth was additionally applied to Schaedler-KV agar with 5% sheep blood and vitamin K1 and incubated under anaerobic conditions with 10% CO2, as described previously, in order to detect anaerobic bacteria [47]. Gram staining was performed on all samples. To assess the presence of remaining antibiotics in samples, which may contribute to false negative results, one aliquot was incubated on B. subtilis agar in order to detect antimicrobial activity in specimens. In positive cases, colonies were picked and analyzed in MALDI TOF MS (Biomerieux Vitek MS, Nürtingen, Germany) for strain identification.

Antibiotic susceptibility testing was performed in the Vitek 2 analyzer (Biomerieux, Nuertingen, Germany) and, in the case of reserve antibiotics, was supplemented by breakpoint analysis or, in single cases, by disk diffusion. Antibiotic susceptibility was evaluated according to EUCAST regulations and defined as susceptible (S), intermediate (I; susceptible with higher antibiotic dose), and resistant (R) [48].

The results were analyzed according to Atkins et al. [45], Pandey et al. [37], Virolainen et al. [43], the criteria of the Musculoskeletal Infection Society (MSIS) [39], and the ICM-2018-Definition [40] whereby a synovial membrane sample was regarded as positive when at least one of the following conditions had been fulfilled:Identification of the same pathogen in at least two of the samples.Identification of a pathogen in at least one sample and demonstration of at least five neutrophilic polymorph leukocytes in five high power fields (×400) in the associated histological preparation and an elevated CRP-value (>10 mg/L) as described by Feldman et al. [41], in the MSIS-criteria [39] and in the ICM-2018-Definition [40].

Cases of positive histology and elevated CRP-value, but no detected microorganism, were rated as culture-negative infections. The presence of bacteria in only one sample without any histological confirmation was regarded as a result of contamination during the sampling procedure or during the incubation period, in accordance with Virolainen et al. [46].

In the patients with bacterial arthritis, after the 14 days of intravenous antibiotic therapy (Rifampicin additionally oral when indicated), an oral antibiotic was administered for another 4 weeks. In the cases with identified pathogens, antibiotic therapy was targeted specifically to the susceptibility of the bacterium (according to the recommendations of a microbiologist specializing in this field). In patients without detectable bacteria, broad-spectrum antibiotic therapy was carried out with a 2nd generation of cephalosporin (Cefuroxime) (Table 1). Patients were followed up for at least 2 years. The mean follow-up was 65.3 ± 27.1 (24–85) months.

Patients were classified as free from reinfection according to Diaz-Ledezma et al. [34] if they met the following criteria: free from mortality related to PJI, free from subsequent surgical intervention for PJI, microbiological as well as clinical absence of the infection for at least 24 months. The in-house detection threshold for CRP was set at ≥10 mg/L [35]. The clinical results were analyzed using the Knee Society Knee Scoring System (KSS) [12].

IBM SPSS Version 24 (IBM Corp., Aemonk, NY, USA) and Microsoft Excel (Microsoft, Redmond, WA, USA) were used for statistical analysis. Categorical variables are depicted as frequencies, while continuous variables are shown as medians, standard deviations, and ranges.

## 5. Conclusions

Unexpected intraoperative bacterial synovitis is a very rare situation with an incidence of 0.12%. In such cases, the use of bone cement with two antibiotics (Copal G+C in this study) and the immediate initiation of systemic antibiotic therapy seem to keep the risk of periprosthetic infection at the same level as in normal gonarthrosis. However, this does not mean that the direct implantation of knee prostheses should be undertaken in cases of proven bacterial arthritis. In this case, treatment of the bacterial arthritis is required first. The incidence of unexpected bacterial arthritis should also be kept as low as possible by carrying out exact preoperative diagnostic procedures with the aspiration of the joint if inflammation parameters are elevated.

## Figures and Tables

**Table 1 antibiotics-12-01153-t001:** Data of the nine patients of this study, RA = rheumatoid arthritis, MTX = methotrexate, Art. = arterial, CRP = C-reactive protein, HPF = high power field, KSS = Knee Society Score.

Gender	Age	History	CRP Preop mg/L	Leuco Preop n/µL	Synno-Vits Type Gächter and Stutz [4]	Histo-Logy	Culture	Antibiotic First 2 Weeks.	Antibiotic Weeks 3 to 6	Follow-Up Months	Clinical Outcome 2 Years Postop
Male	71	Injection Hyaluron-acid (3 months before) Gastritis Diabetes mellitus type II	14.1	6000	Type III	15/HPF	*Cutibacterium* *acnes* in 5 of 5 samples	Penicillin G 3 × 5 million units + Rifampicin 600 mg 0-1-0	Levofloxacin 500 mg 1-0-1 + Rifampicin 600 mg 0-1-0	78	KSS 90
Male	84	RA (MTX) Renal cell carcinoma	50.1	5900	Type IV	20/HPF		Cefuroxim 2 g 1-1-1	Levofloxacin 500 mg 1-0-1	83	KSS 75
Male	55	Injection Hyaluron-acid (2 ½ months before) Art. Hypertension Gastritis	12.5	10,900	Type III	20/HPF	*Staph. saprophyticus* in 3 of 5 samples	Flucloxacillin 4 g 1-1-1 + Rifampicin 600 mg 0-1-0	Levofloxacin 500 mg 1-0-1 + Rifampicin 600 mg 0-1-0	78	KSS 100
Male	58	Arthroscopy (6 months before) hypothyroidism	22.8	9700	Type III	27/HPF		Cefuroxim 2 g 1-1-1	Levofloxacin 500 mg 1-0-1	72	KSS 95
Female	66	Uterine carcinoma Thyroid carcinoma	29.0	4390	Type III	25/HPF		Cefuroxim 2 g 1-1-1	Levofloxacin 500 mg 1-0-1	24	KSS 90
Male	74	RA, osteosynthesis distal femur fracture (2 years before) Coronary heart disease	24.5	5300	Type IV	20/HPF		Cefuroxim 2 g 1-1-1	Levofloxacin 500 mg 1-0-1	85	KSS 75
Male	68	Injection Hyaluron-acid (3 months before)	22.8	8500	Type III	100/HPF		Cefuroxim 2 g 1-1-1	Levofloxacin 500 mg 1-0-1	83	KSS 95
Female	87	Art. Hypertension	25.0	5160	Type III	100/HPF		Cefuroxim 2 g 1-1-1	Levofloxacin 500 mg 1-0-1	24	KSS 70
Female	62	Arthroscopy (7 months before) Diabetes mellitus type II	11.8	8310	Type III	50/HPF		Cefuroxim 2 g 1-1-1	Levofloxacin 500 mg 1-0-1	26	KSS 85

**Table 2 antibiotics-12-01153-t002:** Antibiogram of the two patients with bacterial arthritis and detected microorganism, + = sensibel, − = resistent.

	Patient 1	Patient 3
	Microorganism	Microorganism
Antibiotic	*Cutibacterium acnes* 5 of 5 samples	*Staph. saprophyticus* 3 of 5 samples
Penicillin	+	−
Oxacillin		+
Ampicillin		
Piperacillin		
Cefaclor		
Cefuroxim		+
Cefazolin	+	
Cefotaxim		+
Ceftriaxon		+
Imipenem		+
Meropenem		+
Gentamicin		+
Clindamycin	+	+
Ciprofloxacin		+
Levofloxacin		+
Moxifloxacin	+	+
Rifampicin	+	+
Fusidinsäure		−
Tigecyclin		+
Vancomycin		+
Teicoplanin		+
Linezolid		+
Fosfomycin		−
Daptomycin		+

## Data Availability

We do not wish to share our data because some of the patients’ data regarding individual privacy, and according to the policy of our hospital, the data cannot be shared with others without permission.

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
