# Peer review of "Total Knee Arthroplasty in Unrecognized Septic Arthritis—A Descriptive Case Series Study"

_antibiotics, 2023, doi:10.3390/antibiotics12071153_

Round 1
Reviewer 1 Report
The aim of Authors was to analyse cases of synovitis during endoprotheses for incidence, detection of bacteria, risk factors and outcome.
The manuscript represents an overview of internal data and the incidence of some "unexpected" cases of arthritis during surgery.
The Authors should:
1) carefully explain why these cases are unexpected
2) correlate some risk factors with this by incuding also a probable microbial contamination
3) improve microbiological methods
4) improve the name of microorganisms and chemical compounds
Many mistakes and errors are present along the text
Author Response
The answer is in the added word file

Reviewer 2 Report
The flow of the introduction lacks smoothness and fails to elaborate on the aim of the study. The introduction should effectively convey the necessary background knowledge to enable readers to comprehend the study. Although the introduction provides some background information, the connection between the existing knowledge and the study at hand is not clearly articulated.
The five questions mentioned in the introduction (lines 55-60) should be described as descriptive goals in the introduction and methods sections or removed altogether.
Table 1: should include footnotes for all abbreviations.
Table 1 should include the dosage and duration of treatment (i.e., antibiotics) considering the limited number of patients (nine cases). Additionally, the clinical outcome should also be included in Table 1.
The results section oversimplifies the findings, providing insufficient details for each case. Since only nine cases were reported, it is crucial to provide a comprehensive description of the key findings and their clinical significance.
Regarding line 63, "Of the 25 suspected synovitis cases in the 7534 knee replacements (0.33%), 9 (0.12%) were classified as bacterial (Table 1)," it is essential to clarify whether this search was conducted in a single-centre or multi-centre setting. As the Methods section appears at the end of the paper, it is challenging to understand the nature of the database used. Please verify and provide specific details.
In line 68, "The risk factors for periprosthetic infection in the patients were rheumatoid arthritis in 2 cases and cancer in 1 case (Table 1)," further analysis is needed to support this statement. It is not appropriate to draw conclusions based on N=1 or N=2 cases in this study. I suggest rephrasing this statement or moving it to the discussion section. While I do not disagree with the authors, it is necessary to conduct further studies despite what the literature says. Additionally, statistical analysis should be performed to identify potential risk factors. Since the aim of the study was not clearly identified in the introduction, it is unclear if this result holds significance.
In the discussion section (line 79), the statement "address total knee arthroplasty when unexpected bacterial arthritis is encountered intraoperatively" requires references to support its validity.
More details are needed to support the statement in line 71, "There were no cases of periprosthetic infection of the total knee prosthesis nor other complications within the minimum follow-up of 2 years."
Regarding line 82, "It is striking that 8 of these 9 patients had had injections into the corresponding knee joint, previous operations, or an underlying immune-modulating disease," it is important to explore the existing literature and provide references to explain this phenomenon. One might argue that all interventions and invasive procedures increase the risk of infection.
Line 85 states, "the incidence of septic arthritis is approximately 10-fold higher in patients with RA." It is essential to clarify whether this association is solely linked to RA or if it is also associated with immunosuppressants and corticosteroid use in RA and rheumatology patients. This point should be clarified.
In line 96, the statement "Most of these infections remained culture-negative with no evidence of a microorganism (7 out of 9). In our opinion, this suggests subacute or chronic infections with slow-growing and difficult-to-cultivate bacteria" should be rephrased or supported by references. It is not necessarily true that the difficulty in culturing the bacteria indicates subacute or chronic infections unless Staphylococcus aureus is involved. Several factors can affect the success rate of bacteria culture.
Line 127 mentions, "However, the relatively high number of neutrophil leukocytes of 20 per high-power field in the two cases with rheumatoid arthritis in our study also suggested, in our opinion, a subacute or previous bacterial infection of the joint.” It would be helpful to include the cell counts in Table 1 to support this statement.
Line 139 states, "Of these, bacterial arthritis was suspected intraoperatively in 25 cases (0.33%) because of the macroscopic appearance of the synovitis." It is unclear what clinical definition or inclusion criteria were used to determine bacterial arthritis during the search. Please provide more details on this matter.
In line 147, it is mentioned, "Five samples from the synovitis were obtained intraoperatively for bacteriological cultivation, and the entire remaining synovitis was then sent for histological examination." In that case, including histology images/data would be a valuable addition to the results and discussion sections.
Line 159 states, "Microorganisms were identified by standard microbiological procedures, including biochemical characterization with the API system (BioMerieux, Nuertingen, Germany) in cases of anaerobic strains or anaerobic bacteria. Antibiotic susceptibility testing was performed by disk diffusion or dilution methods according to the Clinical and Laboratory Standards Institute (CLSI) guidelines." It is advisable for the authors to include these findings in the results section. Additionally, providing bacterial staining figures would enhance the presentation of the results.
Moderate editing is required.
Author Response
The answers are in the added word file

Reviewer 3 Report
The writting of the bacteria especies must be corrected: use italics and the correct specie is Cutibacterium acnes.
The correct name of the machine is Vitek2.
All the test could be improved, the theme is too basic, so, if you want more scientific soundness and significance it necessary to reach a very high quality in the test, in the results and in the conclussions. Try again, you could do it.
Some expression seems to be in a incorrect order.
Please, use the comma much more.
Author Response
The answers are in the added word file

Reviewer 4 Report
no comment
Author Response
The answer is in the added word file

Reviewer 5 Report
Dear authors,
In this paper, you sought to analyse outcomes in synovitic patients with TKA. Please find my comments below.
Title: The title of the paper needs to include the study design. Also, the title itself is not describing the major findings of the study and even more so, it’s unclear what ‘unexpected’ means here. Please consider revising.
Line 16: The time frame you considered to gather your cases needs to be mentioned here.
Line 75: For precision, please specify what data you are referring to. Also, 'agnes' should be 'acnes'.
Line 76: Please spell out RA and HPF.
Line 79: I would advise you avoid commenting on whether this is the first study addressing TKA when unexpected bacterial arthritis is encountered intra-operatively. Likewise, I would advise against using words like ‘striking’ as this judgement needs to be left for the readers to make. Rather, an opening paragraph should be structured more appropriately in the discussion section. First, you should describe why you conducted this study and the clinical importance behind this topic. You then need to describe your main findings and briefly comment on them.
Line 96: Please note this paragraph need to be enhanced so you achieve equal length with the rest of the paragraphs of the discussion section.
Line 92: Is there a particular time frame that needs to be reported here? For instance, is a 3-month window sufficient to safely proceed to TKA after an intra-articular injection? Please clarify. This information also should be added to Table 1.
Line 137: Did you follow any particular reporting guidelines for case series? If yes, then please clarify.
Line 170: The use of the term ‘Demonstration’ seems inappropriate here. Please replace with that of ‘identification’.
Line 193: Statistical analysis section needs to be improved. In particular, I would advise you to report whether or not normality tests were executed. Subsequently you should determine exactly which tests you implemented. Also, p values need to be reported so that the readers are aware of the statistical significance levels.
Minor English language polishing is needed.
Author Response
The answers are in the added word file

Round 2
Reviewer 2 Report
Lines 40-42: You shouldn't leave the paragraph with a question. Suggests changing it to a sentence.
The first paragraph is much improved, but more references are needed. There is only one citation. Also, I suggest you include the clinical definition of unrecognized septic arthritis. My understanding is it refers to a condition where an infection in a joint is not initially diagnosed or detected, leading to a delay in appropriate treatment. Potentially it can be caused by pathogens other than bacteria (e.g., fungi and viruses), but bacterial infections are more common. You might want to suggest what bacteria are the most common with references.
Lines 43-51: I understand this is more of a clinical observation from your experience, but you should be able to find at least one or two references for the treatment duration. Systematic reviews or case reports should be included.
Lines 52-54: You should be able to find a reference for this new statement, but I know it is true.
Lines 65-73: It is uncommon to state your research questions at the end of the introduction. I suggest you put them into sentences (e.g., this study aims to determine (1) the frequency of intraoperatively suspected bacterial arthritis, (2) the number of confirmed cases, (3)...)
Lines 88-89: The footnotes should be at the bottom of the table.
Lines 98-102: Do you think the low rate is due to your limitation (being a single-centre study)? I suggest you state the denominator of the rate (n/N) for clarification. The low-positive rate can also be a limitation.
Lines 170-177: This should be in your results section. Those were your findings, not your methodologies.
Line 198: What other agar plates did you use for the subculture of turbid broths? Please list them so that further studies can repeat them. Also, how did you determine the appropriate agar plates without knowing what pathogens they were? What methodologies did you rely on?
Line 203-204, what were your selection criteria? Based on what factors?
Lines 205-209, where are the susceptibility data not presented? Where can I find these references? What were the significant findings? Was any antimicrobial resistance detected? For the o EUCAST regulations, you would need to add a reference.
Informal English was found to require minor language editing.
Reviewer 3 Report
I believe that everything proposed has been taken into account and its resolution seems appropriate.
Author Response
There is no correction necessary